

# Caries risk assessment by Caries Management by Risk Assessment (CAMBRA) Protocol among the general population of Pakistan–a multicenter analytical study

Azhar Iqbal[1], Yasir Dilshad Siddiqui[2], Farooq Ahmad Chaudhary[3],
Malik Zain ul Abideen[4], Talib Hussain[5], Bilal Arjumand[6],
Mohammed Almuhaiza[7], Mohammed Mustafa[7,8], Osama Khattak[1],
Reham Mohammed Attia[9], Asma Abubaker Rashed[10] and
Sherif Elsayed Sultan[11,12]

[1] Department of Restorative Dentistry, College of Dentistry, Jouf University, Sakaka, Saudi Arabia
[2] Department of Preventive Dentistry, College of Dentistry, Jouf University, Sakaka, Saudi Arabia
[3] School of Dentistry, Shaheed Zulfiqar Ali Bhutto Medical University, Islamabad, Pakistan
[4] Department of Dental Education & Research, College of Dentistry, Bakhtawar Amin Medical & Dental College, Multan, Pakistan
[5] Department of Oral Biology, Women Medical & Dental College, Abbottabad, Pakistan
[6] Department of Conservative Dental Sciences and Endodontics, College of Dentistry, Qassim University, Buraydah, Saudi Arabia
[7] Department of Conservative Dental Sciences, College of Dentistry, Prince Sattam Bin Abdulaziz University, Al-Kharj, Saudi Arabia
[8] Center for Transdisciplinary Research, Saveetha Dental College and Hospital, Department of Conservative Dentistry & Endodontics, Saveetha Institute of Medical and Technical Sciences, Saveetha University, Chennai, Tamil Nadu, India
[9] Conservative Dentistry Department, Faculty of Dentistry, Zagazig University, Zagazig, Egypt
[10] Restorative Dentistry Department, Tanta University, Tanta, Egypt
[11] Department of Fixed Prosthodontics, Tanta University, Tanta, Egypt
[12] Department of Prosthetic Dental Sciences, Jouf University, Sakaka, Saudi Arabia

Corresponding authors
Azhar Iqbal,
dr.azhar.iqbal@jodent.org
Farooq Ahmad Chaudhary,
chaudhary4@hotmail.com

## ABSTRACT

**Background:** Caries risk (CR) assessment tools are used to properly identify individuals with caries risk and to improve preventive procedures and programs. A tool such as CAMBRA determines the precise protective factors of caries and identifies an individual's specific therapeutic intervention. The purpose of this study was to assess the caries risk using the CAMBRA protocol among the general population of Pakistan.

**Methods:** This multicentre analytical study was conducted in ten dental hospitals in different provinces of Pakistan and the caries risk assessment was carried out using a questionnaire that was designed using the Caries Management by Risk Assessment (CAMBRA) protocol. All 521 participants were intra-orally examined to assess oral hygiene status and the presence of disease. Multiple logistic regression test was performed for analysis.

**Results:** A higher number of participants (61.2%) were found to be in the moderate risk category of caries risk assessment. The males are 51% less likely to have caries compared to the females (AOR = 0.49, P = 0.081). The majority of participants (71.3%) had one or more disease indicators, with white spots and visible cavities.

Those with visible, heavy plaque were 13.9 times more likely to have caries compared to those without (AOR = 13.92, $P < 0.001$). Those using calcium and phosphate during the last 6 months were 90% less likely to have caries compared to those not using them (AOR = 0.10, $P < 0.001$). There was no significant interaction between all eight risk factors retained in the final model ($P > 0.05$), the Hosmer and Lemeshow Test $P < 0.001$, classification accuracy = 87.1%, and AUC = 91.2%.

**Conclusion:** The caries risk among the general population of Pakistan is moderate, with significant variation among age groups, education levels, and socioeconomic status.

# INTRODUCTION

Dental disease is unquestionably a public health concern and one of the most common illnesses worldwide, especially dental caries, a condition linked to biofilms (*Yadav & Prakash, 2017*). The World Health Organization estimates that dental caries and cavities affect 60–90% of school-age children and almost 100% of adults globally (*Al-Thani et al., 2018*).

It is a well-known fact that dental caries cannot be controlled by restoration alone, therefore multifaceted strategies and approaches that focus on modifying and correcting oral health behaviors are needed to control this problem (*Brantley et al., 1995*; *Suneja et al., 2017*). The prevention of caries was moved from a surgical model to a medical model with the recent recommendations for early detection and monitoring of caries rather than waiting until a cavity forms, and the percentage of people receiving preventive oral health care has been rising in recent years (*Yon et al., 2019*). By preventing caries, one can maintain a healthy tooth structure, stop enamel from demineralizing, and encourage the body's natural healing processes (*Al-Maliky, Frentzen & Meister, 2020*). With time, the focus of care and treatment in dentistry has changed and more emphasis has been placed on personalized care with a targeted approach and care for the patients, based on the risk that has been advocated. Numerous caries risk (CR) assessment tools are used and tested by oral health care professionals to properly identify individuals with caries risk and to improve preventive procedures and programs. This caries risk assessment targeted approach showed better success in prevention and treatment compared to the conventional identical treatment approach to all patients, independent of their risks (*Khattak et al., 2022*; *Suneja et al., 2017*). The caries risk assessment determines the precise protective factors of caries and identifies an individual's specific therapeutic intervention. Studies have reported disease indicators such as bad oral microbes, absence of saliva, and poor eating habits as determinants of caries. Improving these determinants through planned diet, sealants, saliva stimulators, and fluoride therapy contributes to maintaining the teeth healthy, and these measures can be used to prevent dental caries (*Mejàre et al., 2014*). The scope of patient care can be greatly enhanced by doing CR assessment as it is the foundation of a minimal invasive treatment plan, assisting in proposing the most

appropriate invasive and noninvasive treatment and strategies for the recall (*Mejàre et al., 2014*). Caries Management by Risk Assessment (CAMBRA) developed by the American Dental Association is an evidence-based approach through which the higher caries risk individuals can be identified in the earliest stage (*Featherstone, Crystal & Gomes, 2019*). Dentists identify the caries risk level of the individual by evaluating their disease indicators, risk factors, and preventive factors using a caries risk assessment form. Taking those factors into account, a caries risk level of low, moderate, high, or extreme is assigned. This helps in the early adoption of caries preventive measures and focuses on rectification of causes of dental caries, rather than waiting for irreversible damage to the teeth. With the help of CAMBRA protocol each individual can assess for his or her unique disease indicators, risk factors, and protective factors for dental caries (*Hänsel Petersson, Fure & Bratthall, 2003*; *Iqbal et al., 2022*). Early identification of individuals at high risk of caries and providing them with the preventive and management services is particularly important for undeveloped countries like Pakistan where the resources are limited for oral health (*Featherstone, 2004*). However, the Pakistani general population has never been assessed for caries risk using CAMBRA protocol. Therefore, the purpose of this study was to assess the caries risk using Caries Management by Risk Assessment (CAMBRA) protocol among the general population of Pakistan.

## MATERIALS AND METHODS

This multicenter analytical study was carried out in ten dental hospitals in Punjab (Islamic International Dental Hospital, School of Dentistry Dental Hospital, Shifa Dental Hospital, HITEC-IMS Dental Hospital), Sindh (Ziauddin Hospital, Altamash Dental Hospital, Baqai Dental College Hospital), and KPK (Khyber Dentistry Hospital, Ayub College Of Dentistry Hospital, Rehman College of Dentistry Hospital) provinces of Pakistan using a Caries Risk Assessment (CAMBRA) protocol from 15th January to 30th March 2023. The respondents were recruited using a simple random sampling technique from the general public including patients and their attendees (*e.g.*, patient's family members, relatives, and friends) attending the outpatient department of those dental hospitals. The participants were included if they were above the age of 6 years, understood English, Hindko, Pushto, Punjabi, and Urdu language, and were residents of Pakistan. The Institutional Ethics Review Board of Jouf University gave ethical approval (Reference no. F/JU-96/2023) for this study and the participation was voluntary. Participants were informed about the study, and written consent was taken from them (parent/guardian gave the consent for participants below the age of 12 years). The socio-demographic information was collected which included age, gender, education, profession, and socio-economic status. The caries risk assessment was carried out using a questionnaire, designed by using a practical and predictable CAMBRA caries risk assessment tool, including almost all the factors related to the disease. The CAMBRA tool comprises eight risk and protective factors and four disease indicators. Caries risk factors include those factors that can cause or will cause caries manifestation in the future, caries protective factors include all biological and therapeutic measures that can collectively offset caries risk factors and caries indicators are observations based on its history and activity. The participants were asked to fill in a
questionnaire, then they went through an intra-oral clinical examination and bitewing radiographs were taken for the evaluation of their oral hygiene status and detection of any caries lesions. The individuals were categorized as low risk (no carious lesions, no plaque, optimal fluoride use, and regular dental care); moderate risk (carious lesion in previous 12 months, visible plaque, suboptimal fluoride, and irregular dental care); and high risk (one or more carious lesions, visible plaque, suboptimal fluoride, no dental care, high bacterial challenge, and inadequate saliva flow) (*Iqbal et al., 2022*; *Rechmann et al., 2019*).

Due to financial restraints, some risk factors were excluded such as bacterial count for *S. mutans* and lactobacilli in saliva which required costly complex lab infrastructure and process including skilled workers.

## Statistical analysis

The data was analysed using Statistical Package for the Social Sciences (SPSS IBM, Chicago, IL, USA) version 27. The percentages, frequency, mean, and standard deviation were obtained in the descriptive analysis. To find the significant factors associated with caries (*i.e.*, moderate, high, and extremely high) logistic regression test was carried out. Simple logistic regression analysis was initially performed to obtain the crude odds ratio (COR), and those that had a *P*-value of less than 0.25 were considered important predictors of caries and included in the multiple logistic regression analysis to obtain their adjusted odds ratio (AOR). Variable were selected using forward LR and back LR methods, and the Enter method was run to obtain the final model.

## RESULTS

The socio-demographic characteristics of the participants are presented in Table 1. The study involved 521 patients and a higher number of participants (61.2%) were found to be in the moderate risk category and only 13.4%, were in the 'low' category of caries risk assessment. The mean age of the participants was 32.6 (SD = 13.13), with 45.9% males and 54.1% females. The bachelor's degree (26.1%) was the most reported educational level of the participants and another type of profession reported (41.7%). Furthermore, most of the patients were in the middle class (68.4%).

Table 2 presents the association between general characteristics and caries risk. The results showed that eight factors (*i.e.*, gender, education, visible heavy plaque, deep pits, and fissures, living in a fluoride community, using fluoride toothpaste at least once a day, fluoride varnish in the last 6 months, and using calcium and phosphate paste for at least 6 months) were retained in the final model and therefore considered significant predictors of caries. The males were 51% less likely to have caries compared to the females (AOR = 0.49, *P* = 0.081). For education, those with primary education were 84% less likely to have caries than those with none (AOR = 0.16, *P* = 0.008), those with secondary education were 37% more likely to have caries than those with none (AOR = 1.37, *P* = 0.700), those with higher secondary education were 20% more likely to have caries than those with none (AOR = 1.20, *P* = 0.793), those with bachelor's degrees were 3.5 times more likely to have caries than those with none (AOR = 3.49, *P* = 0.091), those with master's were 3.9 times more likely to have caries than those with none (AOR = 3.92,

**Table 1 Descriptive characteristics of the patients.**

| Variable | Mean (SD) |
|---|---|
| **Age** | 32.6 (13.3) |
| | **N (%)** |
| **Gender** | |
| Male | 239 (45.9) |
| Female | 282 (54.1) |
| **Education** | |
| None | 79 (15.2) |
| Primary | 70 (13.4) |
| Secondary | 126 (24.2) |
| Higher secondary | 56 (10.7) |
| Bachelor | 136 (26.1) |
| Master | 42 (8.1) |
| PhD | 5 (1.0) |
| Other | 7 (1.3) |
| **Profession** | |
| Business | 25 (4.8) |
| Doctor | 26 (5.0) |
| Engineer | 3 (0.6) |
| Housewife | 4 (0.8) |
| Lawyer | 8 (1.5) |
| Teaching | 76 (14.6) |
| Mechanic | 1 (0.2) |
| Student | 161 (30.9) |
| Others | 217 (41.7) |
| **Socioeconomic status** | |
| Lower | 108 (20.8) |
| Lower middle | 11 (2.1) |
| Middle | 355 (68.4) |
| Upper | 45 (8.7) |
| **Caries assessment** | |
| Low | 69 (13.4) |
| Moderate | 315 (61.2) |
| High | 121 (25.4) |

$P = 0.122$), and those with PhD were 73% less likely to have caries than those with none (AOR = 0.27, $P = 0.609$).

Additionally, those with visible, heavy plaque were 13.9 times more likely to have caries compared to those without (AOR = 13.92, $P < 0.001$). Those with deep pits and fissures were 3.2 times more likely to have caries compared to those without (AOR = 3.16, $P = 0.005$). Those living in fluoridated communities were 72% less likely to have caries compared to those living in non-fluoridated communities (AOR = 0.28, $P < 0.003$). Those using fluoride toothpaste at least once daily were 77% less likely to have caries compared to those not using it (AOR = 0.23, $P = 0.005$). Those with fluoride varnish in the last 6 months were 80% less likely to have caries compared to those without (AOR = 0.20, $P = 0.003$). Those using calcium and phosphate during the last 6 months were 90% less likely to have caries compared to those not using them (AOR = 0.10, $P < 0.001$).

**Table 2** The subject's general characteristics and risk of caries.

| Variable | COR [95% CI] | P | AOR [95% CI] | P |
|---|---|---|---|---|
| **Age** | 1.05 [1.03–1.08] | <0.001 | – | – |
| **Gender** | | | | |
| Male | 0.70 [0.42–1.17] | 0.175 | 0.49 [0.22–1.09] | 0.081 |
| Female | 1 | | 1 | |
| **Education** | | | | |
| None | 1 | | 1 | |
| Primary | 0.14 [0.06–0.36] | <0.001 | 0.16 [0.04–0.62] | 0.008 |
| Secondary | 1.01 [0.37–2.72] | 0.719 | 1.37 [0.27–6.90] | 0.700 |
| Higher secondary | 0.81 [0.26–2.56] | 0.988 | 1.20 [0.31–4.56] | 0.793 |
| Bachelor | 0.99 [0.37–2.63] | 0.981 | 3.49 [0.82–14.87] | 0.091 |
| Master | 0.92 [0.25–3.36] | 0.904 | 3.92 [0.70–22.04] | 0.122 |
| PhD | 0.24 [0.04–1.49] | 0.126 | 0.27 [0.02–39.87] | 0.609 |
| **Socio-economic status** | | | | |
| Lower | 1 | | | |
| Lower middle | – | | | |
| Middle | 0.18 [0.06–0.60] | 0.005 | – | – |
| Upper | 0.04 [0.01–0.15] | <0.001 | – | – |
| **Visible heavy plaque** | | | | |
| No | 1 | | 1 | |
| Yes | 8.53 [4.24–17.16] | <0.001 | 13.92 [5.28–36.72] | <0.001 |
| **Frequent snacks** | | | | |
| No | 1 | | | |
| Yes | 0.64 [0.38–1.07] | 0.090 | – | – |
| **Deep pits and fissures** | | | | |
| No | 1 | | 1 | |
| Yes | 1.66 [0.98–2.80] | 0.058 | 3.16 [1.43–6.98] | 0.005 |
| **Recreational drugs** | | | | |
| No | 1 | | | |
| Yes | 0.90 [0.47–1.73] | 0.748 | – | – |
| **Inadequate saliva flow** | | | | |
| No | 1 | | | |
| Yes | 1.07 [0.54–2.14] | 0.846 | – | – |
| **Saliva reducing factor** | | | | |
| No | 1 | | | |
| Yes | 0.53 [0.30–0.96] | 0.035 | – | – |
| **Type of saliva reducing factor** | | | | |
| No | 1 | | | |
| Medication | 1.12 [0.59–2.13] | 0.728 | – | – |
| Radiation | 0.83 [0.18–3.85] | 0.808 | – | – |
| Systemic | 0.45 [0.24–0.87] | 0.017 | – | – |
| **Exposed roots** | | | | |

| Table 2 (continued) | | | | |
|---|---|---|---|---|
| Variable | COR [95% CI] | P | AOR [95% CI] | P |
| No | 1 | | | |
| Yes | 1.47 [0.74–2.92] | 0.270 | – | – |
| **Orthodontic appliances** | | | | |
| No | 1 | | | |
| Yes | 0.26 [0.13–0.52] | <0.001 | – | – |
| **Lives in a fluoridated community** | | | | |
| No | 1 | | 1 | |
| Yes | 0.42 [0.25–0.71] | 0.001 | 0.28 [0.12–0.64] | 0.003 |
| **Fluoride toothpaste at least once daily** | | | | |
| No | 1 | | 1 | |
| Yes | 0.61 [0.32–1.16] | 0.131 | 0.23 [0.08–0.64] | 0.005 |
| **Fluoride toothpaste at least 2x daily** | | | | |
| No | 1 | | | |
| Yes | 0.23 [0.13–0.41] | <0.001 | – | – |
| **Fluoride mouth rinse daily** | | | | |
| No | 1 | | | |
| Yes | 0.13 [0.07–0.22] | <0.001 | – | – |
| **Fluoride varnish in last 6 months** | | | | |
| No | 1 | | 1 | |
| Yes | 0.15 [0.07–0.29] | <0.001 | 0.20 [0.07–0.58] | 0.003 |
| **Chlorhexidine prescribed** | | | | |
| No | 1 | | | |
| Yes | 0.26 [0.14–0.50] | <0.001 | – | – |
| **Xylitol gum/lozenges 4x daily last 6 months** | | | | |
| No | 1 | | | |
| Yes | 0.28 [0.16–0.51] | <0.001 | – | – |
| **Calcium and phosphate paste during last 6 months** | | | | |
| No | 1 | | 1 | |
| Yes | 0.10 [0.06–0.19] | <0.001 | 0.10 [0.05–0.22] | <0.001 |

Finally, the model fit was assessed. There was no significant interaction between all 8 risk factors retained in the final model ($P > 0.05$), the Hosmer and Lemeshow Test $P < 0.001$, classification accuracy = 87.1%, and AUC = 91.2% (Fig. 1).

Figure 1 shows the receiver operating characteristics (ROC) curve of the final logistic regression model of caries with gender, education, visible heavy plaque, deep pits, and fissures, living in a fluoride community, using fluoride toothpaste at least once a day, fluoride varnish in the last 6 months, and using calcium and phosphate paste for at least 6 months.

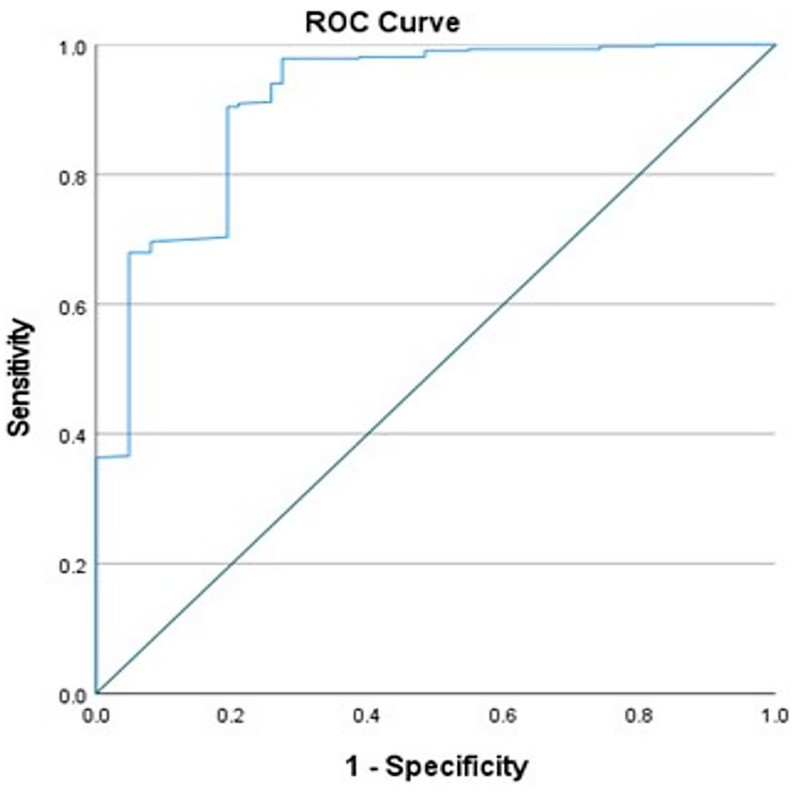

**Figure 1 Receiver operating characteristics (ROC) curve of the final logistic regression model of caries with gender, education, visible heavy plaque, deep pits, and fissures, living in a fluoride community, using fluoride toothpaste at least once a day.**

## DISCUSSION

In this research, the caries risk assessment of the Pakistani population was carried out using caries management by risk assessment protocol. A comparable proportion of male and female subjects were noted in this study. The descriptive characteristic of the study revealed that among 521 patients, 61.2% were judged to have a moderate risk of developing dental caries, whereas 13.4%, 22.7%, and 2.7% were found to be low, high, and at extremely high risk of developing dental caries. Further, logistic regression analysis revealed that eight factors (*i.e.*, gender, education, visible heavy plaque, deep pits, and fissures, living in a fluoridated community, using fluoride toothpaste at least once a day, fluoride varnish in the last 6 months, using calcium and phosphate paste for at least 6 months were considered significant predictors of caries. The result of our investigation revealed that more than half of the participants had restorations within the last 3 years period, and 70% of participants had dentin or enamel lesions that were confirmed clinically and radiographically. According to these facts, dental caries is a risk for the general population, and prompt treatment could avoid worse consequences (*Muhson et al., 2020*). This will not only have positive psychological effects but also have positive financial effects.

Participants for this study were recruited from the outpatient departments of ten dental hospitals in Pakistan. Low levels of water fluoridation were found in most Pakistani provinces, along with high sugar consumption and easy access to sugary foods in public places like schools and offices, which may be one of the probable contributing factors to the nation's higher-than-average dental caries rate (*Chaudhary & Ahmad, 2021*; *Chaudhary et al., 2019*; *Siddiqui et al., 2021*). According to data from a meta-analysis, the prevalence of dental caries in Pakistan's general population was about 60% (*Siddiqui et al., 2021*). Although ages from 6–>60 years were considered in this study, the majority of the participants were between 20 to 39 years group revealing a higher percentage of patients in the moderate risk category, while participants from other age groups were in the low or high-caries-risk groups. These results are comparable to other investigations, where a majority of participants were between 6 to 29 years old (*Gao et al., 2010*; *Mu, 2019*). The comparable proportions of male and female participants were noted in this study with males being 51% less likely to have caries compared to females (*Du et al., 2017*). This is consistent with other studies, where women were in the majority compared to men when identifying the caries risk using CAMBRA (*da Silva Jorge et al., 2021*; *Mu, 2019*; *Siddiqui et al., 2021*). However, this is in contrast to our past research, which involved a Saudi population, where men are at higher risk than women for developing dental caries. The results of this investigation showed that those with heavy plaque were 13.9 times (AOR = 13.92, $P < 0.001$) more likely to have caries compared to those without. Similarly, those having deep pits and fissures were 3.2 times (AOR = 3.16, $P = 0.005$) more along with the frequency of food intake are more prone to developing dental caries. In comparison, categories such as recreational drug use, orthodontic appliances, insufficient salivary flow, exposed roots, and saliva-lowering causes were less common. These observations are similar to the previous studies demonstrating the risk of caries and their relationship with plaque, deep pits and fissures, and frequency of food intake (*Almusawi et al., 2018*; *Bangash et al., 2023*; *Chaffee, Cheng & Featherstone, 2015*; *Iqbal et al., 2022*).

It was observed that participants utilizing fluoride toothpaste and varnish were 77% & 80% less likely to develop dental caries. Additionally, those using calcium and phosphate in the preceding half year had a 90% reduced risk of developing dental caries. These study observations are comparable to prior studies conducted on the Saudi population using CAMBRA (*Iqbal et al., 2022*). Dental caries is substantially less prevalent in modern nations due to better living conditions, rising health consciousness, the use of fluorinated products, and preventive oral care programs (*Muhson et al., 2020*; *Trottini et al., 2015*). In a clinical trial conducted by *Featherstone et al. (2012)* they found that fluoride therapy and targeted antibacterials significantly decreased the degree of caries risk in the intervention groups. The majority of the participants in this study had a moderate risk of getting caries. These findings were in line with several past studies where the moderate-risk group was predominated (*da Silva Jorge et al., 2021*; *Featherstone et al., 2012*).

In a prior study, *Mu (2019)* evaluated the risk of caries in the general population of Lahore, Pakistan, using an American Dental Association (ADA) tool, one of four techniques available for assessing the risk of caries. Contrary to what we found, the majority of people had a high risk of getting caries. In our study, we included more factors

that are strongly related to a lower incidence of dental caries, such as socioeconomic status and education (*Siddiqui et al., 2021*). In terms of education, those with primary education had an 84% lower risk of developing cavities than those without it (AOR = 0.16, *P* = 0.008), those with secondary education had a 37 percent higher risk and those with higher secondary education had a 20 percent higher risk (AOR = 1.20, *P* = 0.793), and those with bachelor's degrees had a 3.5 percent higher risk than those without it (AOR = 3.49, *P* = 0.091). These results are in line with *Almusawi et al. (2018)* study which reported higher caries risk in individuals with more than a primary school level of education. On the other hand in children, the education of the parents was reported as an important factor and associated with a lower risk of caries among their children (*Al-Shammery, 1999*).

When it comes to this study design, there are a few limitations. Primarily, only the general population subjects from dental hospitals were included, which may influence the resulting outcome as the majority showed moderate risk. Additionally, there was no control group for comparison. Moreover, not all the socioeconomic indicators which have been linked to caries risk were considered and bacterial count for *S. mutans* and lactobacilli in saliva were excluded due to financial restraints. Finally, the study was conducted through self-reporting, which leads to inherent response bias.

## CONCLUSION

The result of this study showed that the caries risk identified by CAMBRA among the general population of Pakistan appears moderate. Factors such as age, education level, and socioeconomic status are of importance in the prediction of caries. Daily brushing of teeth with fluoride toothpaste is a great way to reduce caries risk. CAMBRA, a user-friendly tool, can be used to accurately identify the risk factors and create a tailored treatment plan for the patient.

### Funding
The authors received no funding for this work.

### Competing Interests
The authors declare that they have no competing interests.

### Author Contributions
- Azhar Iqbal conceived and designed the experiments, prepared figures and/or tables, and approved the final draft.
- Yasir Dilshad Siddiqui conceived and designed the experiments, prepared figures and/or tables, and approved the final draft.
- Farooq Ahmad Chaudhary conceived and designed the experiments, analyzed the data, prepared figures and/or tables, and approved the final draft.
- Malik Zain ul Abideen conceived and designed the experiments, prepared figures and/or tables, and approved the final draft.

- Talib Hussain conceived and designed the experiments, prepared figures and/or tables, and approved the final draft.
- Bilal Arjumand performed the experiments, prepared figures and/or tables, and approved the final draft.
- Mohammed Almuhaiza performed the experiments, prepared figures and/or tables, and approved the final draft.
- Mohammed Mustafa performed the experiments, authored or reviewed drafts of the article, and approved the final draft.
- Osama Khattak performed the experiments, prepared figures and/or tables, authored or reviewed drafts of the article, and approved the final draft.
- Reham Mohammed Attia performed the experiments, analyzed the data, authored or reviewed drafts of the article, and approved the final draft.
- Asma Abubaker Rashed analyzed the data, authored or reviewed drafts of the article, and approved the final draft.
- Sherif Elsayed Sultan analyzed the data, authored or reviewed drafts of the article, and approved the final draft.

### Human Ethics

The following information was supplied relating to ethical approvals (*i.e.*, approving body and any reference numbers):

The Institutional ethic review board of Jouf University gave ethical approval (Reference no. F/JU-96/2023).

### Data Availability

The raw data is available in the Supplemental File.

### Supplemental Information

Supplemental information for this article can be found online at http://dx.doi.org/10.7717/peerj.16863#supplemental-information.

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
