# Peer review of "Caries risk assessment by Caries Management by Risk Assessment (CAMBRA) Protocol among the general population of Pakistan–a multicenter analytical study"

_PeerJ, doi:10.7717/peerj.16863_

## Round 0.1 · original submission · Major Revisions

Dear authors,

Following the reviewer's comments, I am returning the manuscript to you for suggested modifications and improvements.

Best regards,
CM

Reviewer 1 ·

Basic reporting

no comment

Experimental design

no comment

Validity of the findings

no comment

Additional comments

I have carefully reviewed the manuscript titled "Caries Risk Assessment by Caries Management by Risk Assessment (CAMBRA) Protocol Among the General Population of Pakistan," and I believe that this study makes a valuable contribution to the field of caries risk assessment in Pakistan. The research methodology is generally sound, and the results offer important insights. However, there are a few minor revisions that should be addressed before publication.

Strengths of the Manuscript:
- The study uses a multicenter approach, involving ten dental hospitals across different provinces of Pakistan, which enhances the representativeness of the findings.
- The use of the Caries Management by Risk Assessment (CAMBRA) protocol for caries risk assessment is appropriate and adds to the existing body of knowledge.
- The analysis of risk factors, including gender, oral hygiene status, and the use of calcium and phosphate, provides important insights into caries risk in the Pakistani population.

Minor Revisions Required:
1 - Some sections of the manuscript could benefit from improved clarity and organization. I recommend the authors carefully review and revise the introduction and discussion sections for improved flow and coherence.
2 - In the results section, it would be helpful to provide more context and explanation for the statistical findings, particularly for readers who may not be familiar with the CAMBRA protocol.
3 - Minor typographical and grammatical errors should be addressed to improve overall readability.

Recommendation:
I recommend accepting this manuscript for publication with the condition that the authors address the minor revisions mentioned above. Once these revisions are made, the manuscript will be well-positioned to make a valuable contribution to the field of caries risk assessment and oral health in Pakistan.

·

Basic reporting

The clarity of the English language employed in the text needs to be improved by a fluent English-speaker. The references were comprehensive, providing background information and context within the relevant field. The article was structured professionally, incorporating figures and tables as necessary. Additionally, raw data was shared as part of the content. The article presented results that were pertinent to the hypotheses under consideration.

Experimental design

No comment

Validity of the findings

No comment

Reviewer 3 ·

Basic reporting

It would be extremely important to pay attention to more current references so that the present study is undoubtedly well situated in the literature in the face of contemporary scientific knowledge, even the definition of caries disease is outdated, using a reference from 2006. Also, when describing the etiological factors of the Keyes Triade, it’s important to make reference to the fourth factor, such as time, where the frequency of the other factors was added by Newbrun in 1983.

Experimental design

Although the subject seems interesting, since it was the first time that the caries risk was assessed by Caries Management by Risk Assessment (CAMBRA) protocol in Pakistan, however, some considerations need to be observed:
- Consider adding the CAMBRA Protocol guide, and its reference.
- Which are the 10 hospitals that participated in the research?
- Line 129: describe the choices of the categories, why those? Is there some paper presenting this methodology?
- Despite the financial issues, S. mutans is the most relevant etiological factor for caries diseases, so it is imperative to consider those counts or, at least, describe its importance to the CAMBRA Protocol.

Validity of the findings

It would be interesting to compare those results with up-to-date literature so that this paper can be well positioned to the current understanding.
Furthermore, the fact that the study is using a self-report protocol, also the method of recruitment of the participants, are biased.

Additional comments

I strongly suggest reviewing the proposed methodology to validate your proposal.
Review lines 230 to 235, which seem more like results than a discussion, as well as lines 259 to 266, which seem like conclusions.
It's important to keep in mind the journal you submit your paper, a non-specialized journal has a diverse audience, so make sure you write your manuscript at a level appropriate for the readers of your target journal.

---

## Round 0.2 · accepted · Accept

Dear authors,

Thank you for making the modifications as suggested by the reviewers.
Best regards,

CM